# Multistate Analysis with Infinite Mixtures of Markov Chains

**Lucas Maystre**[1]          **Tiffany Wu**[1]          **Roberto Sanchis-Ojeda**[1]          **Tony Jebara**[1]

[1]Spotify

## Abstract

Driven by applications in clinical medicine and business, we address the problem of modeling trajectories over multiple states. We build on well-known methods from survival analysis and introduce a family of sequence models based on localized Bayesian Markov chains. We develop inference and prediction algorithms, and we apply the model to real-world data, demonstrating favorable empirical results. Our approach provides a practical and effective alternative to plain Markov chains and to existing (finite) mixture models; It retains the simplicity and computational benefits of the former while matching or exceeding the predictive performance of the latter.

## 1 INTRODUCTION

Understanding, modeling and predicting trajectories over multiple states is of central importance in a wide range of applications. For example, in a clinical setting, patients go through several different stages from illness to recovery [Putter et al., 2007]. In a business setting, customers' relationship with a company evolves over time. A customer might start with a free service and later move on to a paid subscription or stop using the service altogether [Pfeifer and Carraway, 2000]. These processes can be viewed as discrete-time or continuous-time sequences over a discrete state space.

In the simplest case, there are only two states and a single transition: Every sequence starts in the first state and ends in the second state. For example, we might be interested in modeling the time a patient takes from admission to a hospital (state 1) to release (state 2). This is the setting of survival analysis, the branch of statistics that studies time-to-event data [Wang et al., 2019]. In this paper, we address a more general setting, where the number of states can be larger than two and the set of admissible transitions can be arbitrary [Cook and Lawless, 2018]. We focus on developing models that accurately capture both the sequence of states and the timing of the transitions. In applications, we use these models to make probabilistic predictions about the future of a sequence given its past.

Markov chains [Norris, 1998] are a popular class of models used to analyze multistate sequences. They come in discrete-time and continuous-time variants, are well-understood theoretically and easy-to-use in practice. One of their strengths is that most problems of interest (learning, prediction, etc.) are tractable, either in closed form or through simple recursive algorithms. However, Markov chains rely on a strong assumption, *memorylessness*, which informally states that future transitions are independent of the past given the present. In practice, this assumption is often too restrictive and can lead to poor predictions. For example, Markov chains are unable to capture the *Lindy effect* [Goldman, 1964], which contends that the longer a process is in a given state, the longer it is expected to remain in that state, and which has been empirically verified in a number of real-world applications [Mandelbrot, 1982, Taleb, 2012].

A common approach to address the limitations of Markov chains is to consider mixtures thereof [Frydman, 1984, Poulsen, 1990, Cadez et al., 2003, Girolami and Kabán, 2003, Frydman, 2005, Gupta et al., 2016]. In short, finite mixture models assume that each sequence follows one of $L \geq 2$ distinct Markov chains. Inference requires explicitly learning the parameters of the $L$ Markov chains and mixture weights associated with each sequence, typically using the EM algorithm [Dempster et al., 1977]. This approach provides increased modeling flexibility but does so the expense of tractability and simplicity. As we show in Section 5.2, running the EM algorithm to convergence requires two orders of magnitude more resources than fitting a single Markov chain. The likelihood function is prone to having poor local maxima, thus necessitating multiple runs with different seeds [Cadez et al., 2003]. These difficulties

*Accepted for the 38th Conference on Uncertainty in Artificial Intelligence* (UAI 2022).

are compounded by the fact that $L$ is usually not known a priori and needs to be chosen and validated empirically.

## 1.1 OUR CONTRIBUTION

In this work, we seek to combine the rich dynamics enabled by mixture models with the convenience and computationally-friendly nature of plain Markov chains. To this end, we develop models of discrete-time and continuous-time sequences based on localized Bayesian Markov chains, following the general construction of Wang and Blei [2018]. Informally, we consider that each sequence follows a latent Markov chain whose matrix of transition rates (in the continuous-time case) or transition probabilities (in the discrete-time case) is sampled from an auxiliary mixing distribution with infinite support (Section 3). We refer to these models as infinite mixtures of Markov chains. The resulting compound process is more expressive than a Markov chain and can capture a wider range of patterns. Furthermore, by choosing the mixing distribution appropriately, the likelihood of a trajectory has a simple closed-form expression, and inference becomes significantly easier than for finite mixtures. We are also able to derive computationally-efficient algorithms for the predictive state distribution (Section 4). Our method can be understood as a generalization of two well-known parametric models used in survival analysis, the beta-logistic and Lomax distributions [Heckman and Willis, 1977, Lomax, 1954], to arbitrary transitions over multiple states.

We evaluate our models empirically on four datasets covering physiological signals, clinical treatment outcomes and customer relationships (Section 5). When, in addition to the sequences themselves, feature vectors are available, our models can be seamlessly combined with regression models. We find that, in each of these datasets, the Markov assumption is too restrictive, and information about a sequence's past helps predicting its future. Our models' predictions outperforms finite mixtures and RNNs, suggesting that the inductive biases of our models are well-suited to these domains. All in all, we believe that our method will be a valuable addition to the practitioner's toolbox.

**A Note on Terminology.** We call our models *infinite* mixtures of Markov chains to emphasize the fact that the (parametric) mixture distribution has infinite support. Our models are distinctly different from nonparametric models such as the infinite Gaussian mixture model [Rasmussen, 1999], the infinite HMM [Beal et al., 2001], and the model of Reubold et al. [2017], which use a Dirichlet process to implicitly capture a variable number of mixture components or latent states.

## 2 RELATED WORK

Sequential data is ubiquitous, and unsurprisingly the literature on models and methods for dealing with such data is vast. Our work addresses applications where the number of states $N$ is finite and typically small with respect to the size of the data, and where accurately modeling the timing of transitions is of particular interest. Correspondingly, we focus our review on the most relevant subset of the literature.

**Survival Analysis.** This field provides the statistical framework for analyzing time-to-event data, i.e., data related to a single transition from one state to another [Klein and Moeschberger, 2003]. Wang et al. [2019] give a recent survey of the field that highlights the connections to machine learning. A popular non-parametric approach to summarizing time-to-event data is given by the Kaplan-Meier estimator [Kaplan and Meier, 1958]. Alternatively, one can postulate a parametric survival distribution and infer the parameters from observed data. The discrete-time beta-logistic model [Heckman and Willis, 1977] and the continuous-time Lomax model [Lomax, 1954] are instances of this approach. The models we develop in this work can be seen as a generalization of these two distributions to multiple states and arbitrary sequences. The beta-logistic model was recently revisited by Hubbard et al. [2021], who report favorable results when used in conjunction with powerful function approximators.

**Multistate Models.** Some methods developed for survival analysis have been extended to handle transitions between $N > 2$ states [Aalen et al., 2008, Putter et al., 2007, Cook and Lawless, 2018]. For example, the Aalen-Johansen estimator generalizes the Kaplan-Meier estimator to trajectories over multiple states [Aalen and Johansen, 1978]. Most methods discussed in the literature are based on a Markov chain model, i.e., they assume that future transitions only depend on the current state. Extensions include time-inhomogeneous or semi-Markov variants, where transition rates can also depend on the absolute time or on the time since the last transition occurred. Fully non-Markovian estimators have recently been proposed [Titman, 2015, Putter and Spitoni, 2018], but they are challenging to use in practice, especially in the small-data regime. Our models are not Markovian—future transitions can depend on the entire history of the process—yet they remain parsimonious, necessitating only twice the number of parameters required to describe a (homogeneous) Markov chain.

**Mixtures of Markov Chains.** The idea of combining multiple Markov chains into a mixture model in order to capture heterogeneity across or within sequences dates back to the 1950s [Blumen et al., 1955]. Frydman [1984, 2005] studies a two-component *mover-stayer* model and its extension to $L \geq 2$ components, with applications to social and finan-

cial processes. Poulsen [1990] and Cadez et al. [2003] use a Markov chain mixture model to cluster customers and users of a website, respectively. Maximum-likelihood inference relies on the EM algorithm [Dempster et al., 1977]. More recently, Gupta et al. [2016] propose an alternative spectral inference algorithm with favorable theoretical properties. Girolami and Kabán [2003] present a different type of Markov chain mixture model where components can be interleaved within a sequence. In contrast to existing work, our approach learns a continuous mixture distribution instead of $L$ discrete components. We compare our models against finite mixture models in Section 5.

**Bayesian Inference for Markov Chains.** Our models make use of mixture distributions that are conjugate for the likelihood functions of Markov chains. Some of these relationships are well-known and have been used for Bayesian inference of Markov chain parameters, such as in MacKay and Bauman Peto [1995] and in Barber [2012, Chapter 23]. In that case, the main goal is to account for the epistemic uncertainty over a single set of parameters due to finite data. Our work is closer in spirit to Wang and Blei [2018], who consider a general framework to transform a classical Bayesian model into a localized one. In our case, a different set of parameters is associated to each sequence, and the Bayesian prior captures *heterogeneity* across sequences. To the best of our knowledge, our work is the first to take advantage of conjugate distributions to learn a mixture of Markov chains.

**Modeling Customer Relationships.** Our work is also related to and influenced by literature on modeling customer retention [Fader and Hardie, 2009]. Schmittlein et al. [1987] estimate time-to-churn by means of a (latent) Lomax survival model. Fader and Hardie [2007] consider a discrete-time variant and use a beta-logistic model. Beyond retention, Pfeifer and Carraway [2000], Paauwe et al. [2007], Schwartz et al. [2011] propose multistate models of customer relationships based on Markov chains. We apply our models to a customer relationship dataset in Section 5.

# 3 STATISTICAL MODELS

In this section, we introduce our sequence models. We begin with a few preliminaries introducing terminology and notation in Section 3.1. Then, we present the discrete-time variant of our method in Section 3.2. We sketch the continuous-time variant in Section 3.3 and link our work to parametric survival models in Section 3.4.

## 3.1 PRELIMINARIES

We consider sequences on $N$ states denoted by the consecutive integers $[N] = \{1, \ldots, N\}$. In discrete time, we define

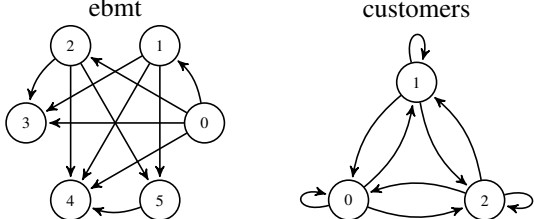

Figure 1: Graph of admissible transitions for the EBMT and CUSTOMERS datasets, analyzed in Section 5.

a sequence of length $T$ as a tuple $s = (s_1, \ldots, s_T)$, where $s_t \in [N]$ for all $t$. In continuous time, we define a sequence over an interval of length $T$ as a function $s : [0, T] \to [N]$, such that $s(t)$ indicates the state at time $t$. In practice, we can represent this function in a compact way by using a discrete sequence of states and the time of each transition. We collect $M$ independent sequences into a dataset $\mathcal{D} = \{s^m : m \in [M]\}$. We allow the length of the sequence (or the length of the interval over which it is defined) $T^m$ to be different for different $m$. In some cases, we associate to each sequence $s^m$ a feature vector $\boldsymbol{x}^m \in \mathbf{R}^D$ that captures additional information about the sequence.

The process generating the sequences is described by a directed graph $\mathcal{G} = ([N], \mathcal{E})$, where the edge set $\mathcal{E} \subseteq [N] \times [N]$ represents the set of admissible transitions. Examples are provided in Figure 1. A state $i \in [N]$ that has no outgoing edges (self-loop excepted) is called *absorbing*. A sequence can—but does not need to—end in an absorbing state. A sequence that does not end in an absorbing state is called *right-censored* [Aalen et al., 2008]. Throughout Section 3, in order to simplify the notation, we assume that all transitions are admissible. However, our developments generalize to arbitrary transition graphs seamlessly, and the applications we study in Section 5 typically only involve a subset of all possible transitions.

Finally, we recall a few well-known functions and distributions. The gamma function is defined as $\Gamma(x) = \int_0^\infty u^{x-1} e^{-u} du$ for $x > 0$. The beta function is defined as $B(\alpha, \beta) = \Gamma(\alpha)\Gamma(\beta)/\Gamma(\alpha + \beta)$. The gamma distribution has support on $\mathbf{R}_{>0}$ and density

$$\text{Gamma}(x \mid \alpha, \beta) = \frac{\beta^\alpha}{\Gamma(\alpha)} x^{\alpha-1} e^{-\beta x},$$

where $\alpha, \beta \in \mathbf{R}_{>0}$ are shape and rate parameters, respectively. The generalized Dirichlet distribution has support on the set of $N$-dimensional probability vectors and density

$$\text{GDir}(\boldsymbol{x} \mid \boldsymbol{\alpha}, \boldsymbol{\beta}) = \prod_{i=1}^{N-1} \frac{x_i^{\alpha_i-1}(1 - x_1 - \cdots - x_i)^{\gamma_i}}{B(\alpha_i, \beta_i)},$$

where $\gamma_i = \beta_i - \alpha_{i+1} - \beta_{i+1}$ for $i = 1, \ldots, N-2$ and $\gamma_{N-1} = \beta_{N-1} - 1$, and $\boldsymbol{\alpha}, \boldsymbol{\beta} \in \mathbf{R}_{>0}^{N-1}$ are parameter vectors.

It extends the Dirichlet distribution by enabling some dependence between the dimensions, and is a conjugate prior to the multinomial distribution [Connor and Mosimann, 1969, Wong, 1998].

## 3.2 DISCRETE-TIME MODEL

We now introduce the discrete-time variant of our model. It builds on homogeneous discrete-time Markov chains (DTMCs), a class of models for sequences that satisfies

$$\mathbf{P}[s_{t+1} = j \mid s_t = i, s_{t-1}, \ldots, s_1] = \theta_{ij}.$$

That is, the probability of transitioning from $s_t$ to $s_{t+1}$ does not depend on the past $s_1, \ldots, s_{t-1}$ (Markov property) nor on the time $t$ (homogeneity). A DTMC is parametrized by the $N^2$ transition probabilities between each pair of states, arranged in the transition matrix $\boldsymbol{\Theta} = [\theta_{ij}]$. Since each row of $\boldsymbol{\Theta}$ sums to one, there are in fact only $N(N-1)$ free parameters. Given a sequence $s$, let $k_{ij} = |\{t : s_t = i, s_{t+1} = j\}|$ count the number of transitions from state $i$ to state $j$. The matrix $\boldsymbol{K} = [k_{ij}]$ is a sufficient statistic for $\boldsymbol{\Theta}$, and the likelihood is given by

$$p(s \mid \boldsymbol{\Theta}) = \prod_{i,j} \theta_{ij}^{k_{ij}}. \tag{1}$$

Given a dataset of sequences, we can find the maximum-likelihood estimate of $\boldsymbol{\Theta}$ by solving a convex optimization problem. The simplicity of DTMCs is appealing, but the Markov property is seldom verified in practice and thus DTMCs can lead to poor predictions.

To overcome this limitation, we proceed as follows. Instead of assuming that all sequences follow the same DTMC, we posit that each sequence follows a *different* DTMC, and we treat the transition matrix $\boldsymbol{\Theta}$ as a latent variable. Furthermore, we posit that, for a given sequence, $\boldsymbol{\Theta}$ is sampled from a product of independent generalized Dirichlet distributions,

$$p(\boldsymbol{\Theta} \mid \boldsymbol{A}, \boldsymbol{B}) = \prod_i \text{GDir}(\boldsymbol{\theta}_i \mid \boldsymbol{\alpha}_i, \boldsymbol{\beta}_i),$$

where $\boldsymbol{A} = [\boldsymbol{\alpha}_i]$ and $\boldsymbol{B} = [\boldsymbol{\beta}_i]$. In other words, each row of $\boldsymbol{\Theta}$ is sampled from a distinct GDir distribution independently of the other rows. We are no longer interested in learning $\boldsymbol{\Theta}$ directly, but instead we seek to learn the parameters of the mixture distribution. Informally, we expect the resulting compound model to be more expressive, since it captures a distribution over infinitely many different DTMCs, as opposed to a single one. Our specific choice of mixture distribution is conjugate for the DTMC likelihood (1). Thus, we can write the compound likelihood (obtained by marginalizing out $\boldsymbol{\Theta}$) in closed form as

$$p(s \mid \boldsymbol{A}, \boldsymbol{B}) = \int p(s \mid \boldsymbol{\Theta}) p(\boldsymbol{\Theta} \mid \boldsymbol{A}, \boldsymbol{B}) d\boldsymbol{\Theta}$$
$$= \prod_{i=1}^{N} \prod_{j=1}^{N-1} \frac{B(\alpha_{ij} + k_{ij}, \beta_{ij} + \sum_{\ell=j+1}^{N} k_{i\ell})}{B(\alpha_{ij}, \beta_{ij})}. \tag{2}$$

Given a dataset of independent sequences $\mathcal{D}$, we can estimate the parameters $\boldsymbol{A}, \boldsymbol{B}$ by minimizing the negative log-likelihood (NLL)

$$\ell(\boldsymbol{A}, \boldsymbol{B}) = -\sum_{s^m \in \mathcal{D}} \log p(s^m \mid \boldsymbol{A}, \boldsymbol{B}). \tag{3}$$

The NLL is not concave in $\boldsymbol{A}$ and $\boldsymbol{B}$, but it has at most one stationary point [Levin and Reeds, 1977], and in practice the maximizer can be found efficiently[1]. Note that the number of free parameters in our model (i.e., in $\boldsymbol{A}, \boldsymbol{B}$) is exactly twice that of a Markov chain (i.e., in $\boldsymbol{\Theta}$).

**Bayesian Update.** Assume that we observe the first $C < T$ steps of a sequence $(s_1, \ldots, s_T)$. What is the likelihood of the second part of the sequence $s' = (s_C, \ldots, s_T)$ given the first part $s = (s_1, \ldots, s_C)$? We can use the conjugacy properties of the mixture distribution to derive

$$p(s' \mid s, \boldsymbol{A}, \boldsymbol{B}) = p(s' \mid \tilde{\boldsymbol{A}}, \tilde{\boldsymbol{B}}),$$

where $\tilde{\boldsymbol{A}} = \boldsymbol{A} + \boldsymbol{U}$ and $\tilde{\boldsymbol{B}} = \boldsymbol{B} + \boldsymbol{V}$ for $\boldsymbol{U}, \boldsymbol{V} \in \mathbf{N}^{N \times (N-1)}$ such that $u_{ij} = k_{ij}$ and $v_{ij} = \sum_{\ell > j} k_{i\ell}$, and $k_{ij}$ counts the number of times the transition $(i, j)$ is observed in the subsequence $s$ [Connor and Mosimann, 1969]. This property highlights that the compound process is not Markovian: The probability of future transitions depends on the entire past of the sequence.

**Combination with Regression Models.** If, in addition to the sequences themselves, we are given feature vectors describing each sequence, we can reparametrize the model by using functions $\boldsymbol{A}(\cdot)$ and $\boldsymbol{B}(\cdot)$ that map feature vectors to positive-valued parameter matrices. This lets us combine our sequence model with any machine-learning regression model. For example, we obtain a log-linear model by setting $\boldsymbol{A}(\boldsymbol{x}) = [\alpha_{ij}(\boldsymbol{x})]$ with $\alpha_{ij}(\boldsymbol{x}) = \exp \boldsymbol{w}_{ij}^\top \boldsymbol{x}$, and likewise for $\boldsymbol{B}(\boldsymbol{x})$. Alternatively, we could use regression trees or deep neural networks, similarly to Hubbard et al. [2021]. Instead of optimizing (3) over matrices $\boldsymbol{A}$ and $\boldsymbol{B}$, we would then optimize over the parameters of the matrix-valued functions $\boldsymbol{A}(\cdot)$ and $\boldsymbol{B}(\cdot)$.

## 3.3 CONTINUOUS-TIME MODEL

The continuous-time version of our model builds on homogeneous continuous-time Markov chains (CTMC). A CTMC is parametrized by the $N \times N$ infinitesimal generator matrix $\boldsymbol{\Lambda} = [\lambda_{ij}]$, where, for every $i \neq j$, $\lambda_{ij} > 0$ is the instantaneous rate of transition from state $i$ to state $j$, and $\lambda_{ii} = -\sum_{j \neq i} \lambda_{ij}$. Given a sequence $s$, let $\boldsymbol{K} = [k_{ij}]$ such that $k_{ij}$ counts the number of transitions from state $i$

---

[1] Most machine-learning frameworks include $\log B(\alpha, \beta)$ as a differentiable primitive. In TensorFlow for example, it is available under `tf.math.lbeta`.

to state $j$, and let $\boldsymbol{\tau} = [\tau_i]$ such that $\tau_i = \int_0^T \mathbf{1}_{\{s(t)=i\}} dt$ is the total time spent in state $i$. Then the pair $(\boldsymbol{K}, \boldsymbol{\tau})$ is a sufficient statistic for $\boldsymbol{\Lambda}$, and the likelihood is given by

$$p(s \mid \boldsymbol{\Lambda}) = \prod_i e^{\lambda_{ii}\tau_i} \prod_{j \neq i} \lambda_{ij}^{k_{ij}}. \qquad (4)$$

Similarly to the discrete-time case, we posit that each sequence follows a different CTMC and treat $\boldsymbol{\Lambda}$ as a latent variable. We assume that each $\lambda_{ij}$ is sampled from a distinct, independent gamma distribution:

$$p(\boldsymbol{\Lambda} \mid \boldsymbol{A}, \boldsymbol{B}) = \prod_{i \neq j} \mathrm{Gamma}(\lambda_{ij} \mid \alpha_{ij}, \beta_{ij}).$$

As in the discrete-time case, the mixture model is described by $2N(N-1)$ free parameters, twice that of a CTMC. The product of Gamma mixture distribution conjugates with the likelihood (4), and the compound likelihood is available in closed form as

$$\begin{aligned} p(s \mid \boldsymbol{A}, \boldsymbol{B}) &= \int p(s \mid \boldsymbol{\Lambda}) p(\boldsymbol{\Lambda} \mid \boldsymbol{A}, \boldsymbol{B}) \\ &= \prod_{i \neq j} \left[ \frac{\Gamma(\alpha_{ij} + k_{ij})}{(\beta_{ij} + \tau_i)^{\alpha_{ij} + k_{ij}}} \cdot \frac{\beta_{ij}^{\alpha_{ij}}}{\Gamma(\alpha_{ij})} \right]. \end{aligned} \qquad (5)$$

In general, the points we made for the discrete-time model in Section 3.2 extend to the continuous-time model consistently. The maximum-likelihood estimate can be found efficiently, the sequence model can be combined with function approximators, and the properties of the compound process are similar in both discrete and continuous-time.

### 3.4   CONNECTION TO SURVIVAL MODELS

We consider the case where $N = 2$, all sequences start in state 1 and state 2 is absorbing. This is the classic setting studied in the survival analysis literature. In the discrete-time case, we can rewrite (2) as

$$p(s \mid \alpha, \beta) = \frac{B(\alpha + k_{11}, \beta + k_{12})}{B(\alpha, \beta)},$$

where $\alpha, \beta > 0$, $k_{11}$ is the number of steps the sequence has remained in state 1 and $k_{12}$ is a binary variable indicating whether state 2 has been reached (i.e., whether the observation is uncensored or right-censored). This is exactly equivalent to the beta-logistic model of Heckman and Willis [1977], also known as the (shifted) beta-geometric distribution.

In the continuous-time case, we can rewrite (5) as

$$p(s \mid \alpha, \beta) = \left( \frac{\alpha}{\beta} \right)^{k_{12}} \left( \frac{\beta}{\beta + \tau_i} \right)^{\alpha},$$

where, similarly, $k_{12}$ can be thought of as a censoring indicator variable. This recovers the Lomax distribution [Lomax, 1954], a special case of Pareto Type-II distribution.

The connection to these survival distribution helps explain the inductive biases of our model. Both the beta-logistic and the Lomax distributions are heavy-tailed, and they can thus capture the *Lindy effect* [Goldman, 1964]: The longer the process stays in state 1, the longer it is expected to stay in state 1. This is in contrast to DTMCs and CTMCs, which, in the setting of survival analysis, correspond to geometric and exponential survival distributions, respectively—both light-tailed, memoryless distributions.

## 4   PREDICTIVE STATE DISTRIBUTION

We now focus on the following problem: given a (trained) discrete-time model, an initial state distribution $\boldsymbol{\pi}_0$ and a time horizon $T$, predict the marginal state distribution after $T$ steps, $\boldsymbol{\pi}_T^\star$. This distribution is the multistate equivalent of the survival distribution in survival analysis, and it is of central importance in many applications. For example, in a disease progression model, it can be used to predict the number of patients that have recovered after a given time, irrespective of the patients' particular trajectories. In the case of Markov chains there is an efficient algorithm to compute the state distribution exactly, with running time $O(N^2 T)$. Given a transition matrix $\boldsymbol{\Theta}$ and starting from $\boldsymbol{\pi}_0^\star = \boldsymbol{\pi}_0$, we can use the identity $\boldsymbol{\pi}_t^\star = \boldsymbol{\pi}_{t-1}^{\star\top} \boldsymbol{\Theta}$ iteratively $T$ times to obtain $\boldsymbol{\pi}_T^\star$. The identity is a consequence of the Markov property. To the best of our understanding there is no similar iterative procedure applicable to our model in the general case.

A special case of practical importance is when the directed graph of admissible transitions has no cycles of length greater than one (in other words, the graph is acyclic except for self-loops). In this case, we can derive a simple iterative procedure with running time quadratic in $T$.

**Proposition 1.** *Let $(\boldsymbol{A}, \boldsymbol{B})$ be any generalized Dirichlet mixture of Markov chains on a graph $\mathcal{G} = ([N], \mathcal{E})$, and let $\boldsymbol{\pi}_0$ be an initial state distribution. If $\mathcal{G}$ has no cycle of length greater than one, then $\boldsymbol{\pi}_T^\star$ can be computed exactly in time $O(T^2 N^2)$.*

We provide an explicit algorithm as well as complete proofs of the results presented in this section in Appendix B. By way of example, the transition graph underpinning the EBMT dataset, depicted in Figure 1, satisfies the condition of the proposition.

In the general case where arbitrary cycles are allowed, it is still possible to compute $\boldsymbol{\pi}_T^\star$ exactly with running time polynomial in $T$, but the algorithm is impractical for all but the simplest cases (see Appendix B). However, the specific structure of our mixture model suggests an effective sampling-based approach. The key observation is that, conditioned on a transition matrix $\boldsymbol{\Theta}$ sampled from the mixture distribution, we can compute the predictive state probability

**Algorithm 1** Predictive state distribution.

---

**Require:** $A, B$, horiz. $T$, init. dist. $\pi_0$, # samples $L$
1: **for** $\ell = 1, \ldots, L$ **do**
2: $\quad \Theta \leftarrow$ sample from $\prod_i \text{GDir}(\theta_i \mid \alpha_i, \beta_i)$
3: $\quad \pi_{\ell,0} \leftarrow \pi_0$
4: $\quad$ **for** $t = 1, \ldots, T$ **do**
5: $\quad\quad \pi_{\ell,t} \leftarrow \pi_{\ell,t-1}^\top \Theta$
6: **return** $\hat{\pi}_T = \frac{1}{L} \sum_\ell \pi_{\ell,T}$

---

after $T$ steps exactly by using the efficient recursive algorithm for Markov chains. Therefore, we propose to estimate the state distribution by averaging samples obtained as follows. First, sample $\Theta$ from the mixture distribution, and then compute the exact state distribution conditional on $\Theta$. We formalize this procedure in Algorithm 1.

A natural question to ask is: How many samples are necessary to achieve a desired level of accuracy? We answer this question with a proposition that provides an upper bound on the sample complexity.

**Proposition 2.** *For any $A, B$, horizon $T$, and initial distribution $\pi_0$, let $\hat{\pi}_T$ be the output of Algorithm 1. Then, for any $\epsilon, \delta > 0$, we have $\mathbf{P}[\|\hat{\pi}_T - \pi_T^\star\| < \varepsilon] > 1 - \delta$, as long as $L > \frac{11}{\varepsilon^2} \log \frac{N+1}{\delta}$.*

A useful viewpoint is to think of our sampling scheme as approximating an infinite mixture model with a finite mixture of $L$ components. In contrast to *learning* a finite mixture model, where large values of $L$ can lead to overfitting, increasing $L$ in Algorithm 1 can only increase the accuracy of the resulting estimate. In Appendix B, we run experiments comparing Algorithm 1 to a naive scheme that directly samples trajectories, instead of sampling $\Theta$ and averaging over all possible sequences as we do. We find that our algorithm is significantly more efficient.

**Continuous-Time Model.** We can adapt the same idea to the continuous-time setting as follows. The exact state distribution of a CTMC at time $T$ is given by $\pi_T^\star = \pi_0^\top e^{T\Lambda}$, where the matrix exponential $e^X$ can be well-approximated by a simple iterative procedure. We adapt lines 3–5 of Algorithm 1 accordingly.

## 5 EXPERIMENTAL EVALUATION

In this section, we evaluate the performance of our models empirically on four real-world datasets. First, we investigate model fit (Section 5.1) and running time (Section 5.2) on all four datasets. Then, we focus on two applications and evaluate our models on state prediction tasks (Section 5.3).

**Datasets.** The datasets we study contain sequences describing sleep patterns (SLEEP [Kneib and Hennerfeind,

2008]), two types of clinical treatments and outcomes (VENTICU [Grundmann et al., 2005], EBMT [Fiocco et al., 2008]), and customers' relationship with the Spotify audio streaming service (CUSTOMERS). The first three datasets contain continuous-time sequences, whereas the last dataset contains discrete-time sequences. The number of states $N$ ranges between 2 and 6, and, for all but the last dataset, the set of admissible transitions $\mathcal{E}$ is a strict subset of all possible transitions. The transition graphs of EBMT and CUSTOMERS are illustrated in Figure 1. A more comprehensive description of each dataset is given in Appendix C.

**Experimental Procedure.** Taking the discrete-time case as example, we proceed as follows. We train our models by estimating the parameter matrices $A, B$ of the generalized Dirichlet mixture distributions. We do so by minimizing the negative (marginal) log-likelihood (3) on a training set. At test time, we make use of the parameters estimated during training to make predictions about each sequence in an independent test set.

**Competing Approaches.** We compare our infinite mixture models against $a$) plain Markov chains (denoted by CTMC or DTMC), $b$) finite mixture models trained using EM, and $c$) variants of RNNLM [Mikolov et al., 2010]. For finite mixtures, we choose the number of components $L$ by cross-validation. For the RNN baseline, we note that our goal is not to find the optimal architecture but rather to anchor our results against a well-known representative of this class of models. While discrete-time RNNs are well established, continuous-time variants are still under active research [see discussion in Rubanova et al., 2019]. For our purposes, we extend the RNNLM to continuous-time sequences as follows. At each step, in addition to transition probabilities, we output a transition rate that is a (learned) function of the RNN's hidden state.

**Features.** The EBMT and CUSTOMERS datasets contain, in addition to the sequences themselves, feature vectors that describe characteristics of patients and customers, respectively. In this case, we can combine sequence models with a regression model. We do so by replacing the fixed parameters of a sequence model (e.g., the Markov chain transition matrix $\Theta$ or the GDir parameters $A, B$) with a learned function of the sequence features. For simplicity, we only consider Markov chains, finite mixtures, and our infinite mixtures in combination with an independent log-linear regression model for each parameter (see Section 3.2).

**Reproducibility.** A software library implementing the models and computational notebooks enabling the reproduction of the results presented in this section are provided online.[2] All but one dataset (CUSTOMERS) is publicly avail-

---

[2]See: `https://github.com/spotify-research/mixmarkov`

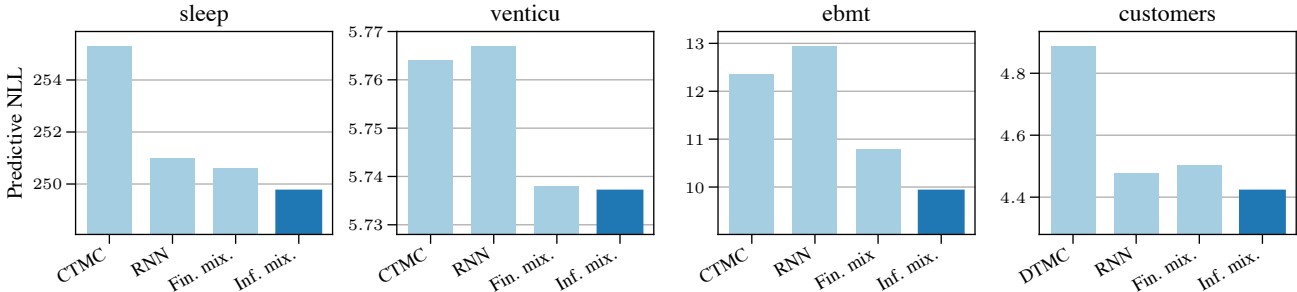

Figure 2: Negative log-likelihood of test sequences under various models on six datasets (lower is better). The three left-most datasets contain continuous-time sequences, the right-most dataset contains discrete-time sequences.

able online. Links to the datasets and additional details on the experimental procedure are provided in Appendix C.

## 5.1 MODEL FIT

We start by reporting the average negative log-likelihood of various models on held-out sequences using 10-fold cross-validation. The NLL provides a consistent and meaningful goodness-of-fit measure for all datasets, irrespective of the application domain. It evaluates the models' ability to jointly predict the identity of the next state and the time until the transition occurs; a lower value corresponds to a better model.

We present results in Figure 2. Our models, highlighted in dark blue, outperform competing approaches on all datasets. Plain Markov chains perform poorly, suggesting that, in all the datasets that we consider, the entire past of a sequence is useful to predict its future (we will revisit this observation in Section 5.3). At the other end of the expressivity spectrum, our results also suggest that RNNs underperform other methods in particular when the dataset is small (SLEEP), or when sequences are short but the timing of transitions is critical (VENTICU, EBMT). Well-tuned finite mixture models perform well, and in some cases they are close to matching the performance of our infinite mixture models (VENTICU).

### 5.1.1 Visualizing Model Fit on CUSTOMERS

The CUSTOMERS dataset represents the trajectories of $144\,510$ users of the Spotify audio streaming service[3] over $N = 3$ states. Users can use the free version of the service (state 1), subscribe and get unrestricted access to all features (state 2), or stop using the service (state 3). A transition can occur between any pair of states (see Figure 1, right). Each sequence starts when the user registers to the service and ends after $T = 20$ steps.

In Figure 3, we visualize the fit of a DTMC and a infinite mixture model. We represent the empirical fraction of pay-

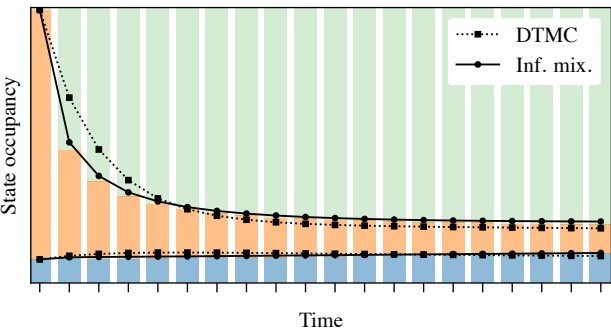

Figure 3: Empirical distribution of users over states (blue, orange and green bars) and predicted distributions, as a function of time.

ing, free and inactive users over time by using blue, orange and green bars, respectively. We indicate the predictive state distribution obtained from the mixture model by using solid lines. Similarly, we use dotted lines to indicate the predictive state distribution obtained from the DTMC. We observe that the mixture model matches the empirical distribution significantly better than the DTMC.[4] Notice how the number of active users (free and paid) decreases steeply after one time step, but then flattens out rapidly. This is a concrete example of the Lindy effect.

## 5.2 RUNNING TIME

Comparing the computational footprint of different models is challenging, as implementation choices can significantly impact results. However, given that Markov chains, finite mixtures and infinite mixtures share many building blocks, we believe that comparing the relative running time of inference in these three models provides insights that will generalize to implementations beyond ours. We use the running time of plain Markov chains as a baseline for each dataset. For finite mixtures of $L$ components, assuming that

---

[3]See: 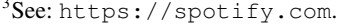 https://spotify.com.

[4]The finite mixture model is not represented in Figure 3, but its fit is also excellent, and almost indistinguishable from that of the infinite mixture model.

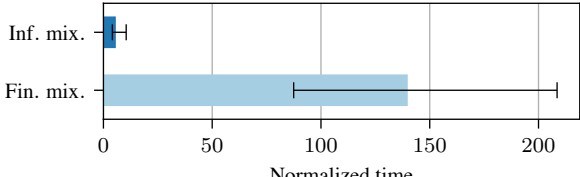

Figure 4: Median and interquartile range for the running time of finite and infinite mixture models, normalized by the running time of a single Markov chain. The median normalized running time is 5.09 and 139.28, respectively.

EM converges in $I$ iterations, parameter inference is dominated by $L \cdot I$ calls to a Markov chain inference subroutine. For infinite mixture models, inference is similar to a Markov chain in that it consists of solving a single, well-behaved optimization problem that can be outsourced to off-the-shelf software.

Figure 4 compares the running time of the two types of mixtures models, normalized by the running time of plain Markov chains and aggregated over all the datasets. We observe that training infinite mixture models takes approximately $27\times$ less time than training finite mixture models. Combined with the predictive edge observed in Figure 2, we believe that this makes our models a compelling alternative to finite mixtures.

## 5.3 PREDICTIVE TASKS

Next, we focus on the EBMT and CUSTOMERS datasets and consider two concrete state predictions tasks.

### 5.3.1 Outcomes in Bone Marrow Transplantations

The EBMT dataset describes patients undergoing bone marrow transplantation, a standard treatment for acute leukemia. The dataset contains trajectories tracking clinical outcomes from the moment the transplantation occurs and spanning up to 18 years. At any time, patients are in one of $N = 6$ states describing the occurrence of adverse events, remission, full recovery, relapse and death. Most patients only go through two or three state transitions. In addition to the trajectory itself, patients are also described by a feature vector encoding demographic and treatment information. The transition graph is depicted in Figure 1 (left), and more details on the data can be found in Fiocco et al. [2008].

We restrict our attention to patients followed over at least 5 years and consider the following task. Given the trajectory of the patient up to day 60, predict the patient's state on day 1800. Being able to accurately estimate the probability of various future outcomes in a personalized way, by using features and recent history, could help identify and follow

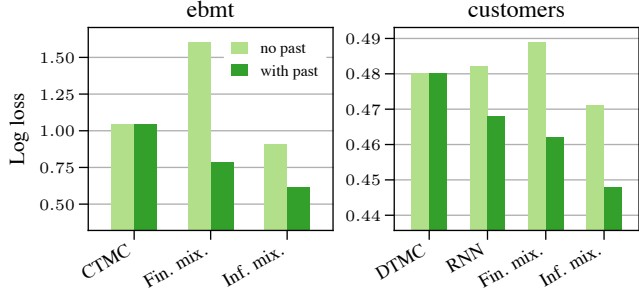

Figure 5: Predictive performance on two state prediction tasks. Predictions are made without and with information about sequences' past (in light and dark green, respectively).

at-risk patients. We train three different models[5] in combination with log-linear regression models and compute the predictive state distribution after $T = 1800$ days on held-out sequences. For each model, the predictive state distribution is computed in two different ways: by using the state on day 60 only, and by using the entire trajectory up to day 60.

We evaluate the prediction by measuring the log-loss given the prediction and the true observed state at the end of the five-year horizon, and we present the results in Figure 5 (left). We observe that using information about the past results in better predictions for models that can take advantage of it, again demonstrating that the Markov assumption is too restrictive. In addition, we observe that the infinite mixture model provides the most accurate predictive state distribution.

### 5.3.2 Modeling Customer Relationships

We set up a similar task on the CUSTOMERS dataset. Given the first 3 time steps of the sequence, we seek to predict the state at step $T = 20$. This task is a multistate extension of the popular problem of estimating customer retention [Fader and Hardie, 2007, Hubbard et al., 2021]. Prior work on modeling complex customer relationships has relied on Markov chains [Pfeifer and Carraway, 2000, Schwartz et al., 2011].

Similarly to the clinical application, we train different sequence models in combination with log-linear regression models, and we compute the predictive state distribution $\pi_T$ on held-out sequences. For each model, we make two predictions: the first one only takes the last observed state $s_3$ into account, whereas the second takes the entire past $(s_1, s_2, s_3)$ into account. The results are presented in Figure 5 (right). Our findings mirror those obtained on the EBMT datasets. Making use of a sequence's past, even if it only consists of three steps, significantly improves the

---

[5] We also experimented with an RNN, but sampling continuous-time trajectories proved difficult and led to poor results.

prediction for all models, and our infinite mixture models outperform competing methods on this task.

# 6 CONCLUSION

We have introduced a new family of models for discrete-time and continuous-time sequences based on infinite mixtures of Markov chains. The models build on principled statistical foundations and extend well-known parametric survival distributions. Our approach retains most of the statistical and computational advantages of plain Markov chains, while enabling predictions whose accuracy matches or exceeds those of finite mixture models.

Our work so far has mostly focused on the prediction problem. In the future, we would like to investigate how our model could help understanding the *mechanisms* driving the sequences. For example, we would like to address questions such as: What is the impact of a given feature on the predicted state distribution? We have preliminary ideas relating our model combined with log-linear regression functions to a Cox proportional-hazards model [Klein and Moeschberger, 2003], and we hope to explore this further.

**Acknowledgements**

We thank Mounia Lalmas, Ksenia Konyushkova and anonymous reviewers for their constructive feedback and careful proofreading.

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
