# OpenReview forum: "Multistate Analysis with Infinite Mixtures of Markov Chains"
_auai.org/UAI/2022/Conference — UAI 2022 Poster_

### Official Review · Reviewer_dkEa · 2022-04-05

**Q2(1) Originality/Novelty:** 3
**Q2(2) Significance/Impact:** 3
**Q2(3) Correctness/Technical Quality:** 4
**Q2(6) Clarity Of Writing:** 4
**Q6 Overall Score:** 7
**Q8 Confidence In Your Score:** 4

**Q1 Summary And Contributions:**

The paper presents a hierarchical Bayesian Markov model for (multiple) time series based on a generalized Dirichlet prior for the transition probabilities. The paper includes a generalization to the continuous time setting. The model is examined and compared with other approaches on several open datasets. The paper is very clear and well written, which makes it easy to follow the ideas. The main technical novelty is somewhat limited.

**Q2 Assessment Of The Paper:**

More detailed information regarding each of these aspects is given below:

**Q2(4) Quality Of Experiments (Optional):**

3: Good: The experimental evaluation is adequate, and the results convincingly support the main claims.

**Q2(5) Reproducibility:**

4: Excellent: Key resources (e.g., proofs, code, data) are available and key details (e.g., proof sketches, experimental setup) are comprehensively described for competent researchers to confidently and easily reproduce the main results.

**Q3 Main Strengths:**

The main idea is simple but solid.
The included software implementation makes it easy to reproduce results.
The paper is clear and includes sufficient technical details.


**Q4 Main Weakness:**

The practical importance of Proposition 2 is not clear, and it is not clear to me from the supplement how well it holds emprically.


**Q5 Detailed Comments To The Authors:**

As you note, the term "infinite mixture" might be confusing. How about using the terms "mixture distribution" and "compound distribution", or perhaps "continuous mixture", or refer to the model as a hierarchical Bayesian model.


**Q7 Justification For Your Score:**

Solid paper. Fair/good technical novelty and good expected significance


**Q9 Complying With Reviewing Instructions:**

1: Yes.

---

### Official Review · Reviewer_ti3i · 2022-04-06

**Q2(1) Originality/Novelty:** 3
**Q2(2) Significance/Impact:** 3
**Q2(3) Correctness/Technical Quality:** 3
**Q2(6) Clarity Of Writing:** 4
**Q6 Overall Score:** 7
**Q8 Confidence In Your Score:** 4

**Q1 Summary And Contributions:**

The authors propose a Bayesian method for modeling trajectories over multiple states in applications where the Markov chain *memorylessness* assumption is unrealistic. Their solution is to create an infinite mixture of Markov chains of different lengths, such that the resulting model is not Markovian, but only requires twice the number of parameters of one chain. They introduce both discrete and continuous-time versions of their model, and employ conjugate priors to ensure tractable computation.

**Q2 Assessment Of The Paper:**

More detailed information regarding each of these aspects is given below:

**Q2(4) Quality Of Experiments (Optional):**

3: Good: The experimental evaluation is adequate, and the results convincingly support the main claims.

**Q2(5) Reproducibility:**

3: Good: Key resources (e.g., proofs, code, data) are available and key details (e.g., proofs, experimental setup) are sufficiently well-described for competent researchers to confidently reproduce the main results.

**Q3 Main Strengths:**

The paper is almost impeccably written and very well structured. It is very easy for the reader to follow the motivation of the authors, the research question they are trying to address, and how it relates to previous work. Their approach of constructing an infinite mixture of Markov chains is straightforward and, to the best of my knowledge, novel. The simplicity of the Bayesian model proposed, as well as its efficiency through the use of conjugate priors, make it likely that the work will have high impact. Finally, the experimental evaluation is detailed and includes relevant examples that truly showcase the kind of applications where this compound model is both necessary and suitable.

**Q4 Main Weakness:**

While the authors do a great job at relating their idea to previous work, I was rather surprised that they did not discuss any other works proposing infinite mixtures of Markov chains. For instance, I would have expected to see their model compared to that proposed in (Reubold et al., 2017, https://link.springer.com/chapter/10.1007/978-3-319-78680-3_12). The work of (Rueckert et al. 2016, https://www.nature.com/articles/srep21142) also seems relevant enough to include in the discussion.

**Q5 Detailed Comments To The Authors:**

On page 5, after Proposition 1, you mention "an explicit algorithm" for computing the predictive state distribution. When first reading this paragraph, I understood that this algorithm was provided in Appendix B, but on a second read I realized you are actually talking about Algorithm 1. Perhaps you could rephrase the sentence a little bit or put a reference to the algorithm in parentheses, so as to avoid any confusion.

For the results presented in Figure 3, you only compared the DTMC to the infinite mixture. It would also be interesting to see how the finite mixture performs in estimating the empirical distribution. Is there any particular reason why the finite mixture was not included in the comparison?

I like that multiple replications were performed, and interquartile ranges were shown for the experiments in Figure 4, but what about for the other experiments? There should be also be some variability in the other results, so it would be nice to get an estimate of this uncertainty.

Minor writing errors:
- Page 2, left column, A Note on Terminology: "distincly" -> 'distinctly'
- Page 4, right column, Bayesian Update: Add 'the' before the very last "sequence".
- Page 8, left column, 5.3.1: "additon" -> 'addition'
- Page 8, right column, 5.3.2: "significiantly" -> 'significantly'
- Page 9, left column, last entry: "sifferent" -> 'different'

**Q7 Justification For Your Score:**

I was impressed by how the authors presented their work in a superbly written paper, in which they managed to emphasize very clearly what kind of applications their model has. The method is simple and efficient, which makes me believe that it will have a significant impact. The only thing really missing is a more ample discussion of Markov chain infinite mixtures, as exemplified by a few references proposing similar models. Nevertheless, there seems to be sufficient novelty in the proposed work.

**Q9 Complying With Reviewing Instructions:**

1: Yes.

---

### Official Review · Reviewer_rF1e · 2022-04-14

**Q2(1) Originality/Novelty:** 3
**Q2(2) Significance/Impact:** 3
**Q2(3) Correctness/Technical Quality:** 4
**Q2(6) Clarity Of Writing:** 4
**Q6 Overall Score:** 6
**Q8 Confidence In Your Score:** 4

**Q1 Summary And Contributions:**

Proposes, analyses, and reports results on making parameters of a Markov chain distinct for each chain and learning the prior over these parameters.

**Q2 Assessment Of The Paper:**

More detailed information regarding each of these aspects is given below:

**Q2(4) Quality Of Experiments (Optional):**

2: Fair: The experimental evaluation is weak: important baselines are missing, or the results do not adequately support the main claims.

**Q2(5) Reproducibility:**

4: Excellent: Key resources (e.g., proofs, code, data) are available and key details (e.g., proof sketches, experimental setup) are comprehensively described for competent researchers to confidently and easily reproduce the main results.

**Q3 Main Strengths:**

+ simple method
+ good related work section, written not not only list, but also explain prior work
+ nice analysis of non-sampling inference method (in appendix)


**Q4 Main Weakness:**

- "learning" a underspecified (see comments below)
- experimental results omit reasonable comparisons

**Q5 Detailed Comments To The Authors:**

I generally liked the simplicity of the method and the clarity of the presentation.  The provided code is clean and easy to read.

I was surprised by a few omissions:

1. If the goal is to allow for non-Markovian state transitions, it would seem natural to compare with a semi-Markov model.  For instance, we could postulate the durations are drawn from any phase-type distribution (thus enabling long tails).  This would seem to be a natural comparison.
2. An HMM would also be a natural comparison, although the "finite mixture" comparison does capture much of this (just an HMM with a hidden state that cannot change).  While the related work section dismisses infinite HMMs and the like quickly, they would still seem to be reasonable comparisons, although certainly more complex than what is presented here.  (while I consider the omission of semi-Markov models -- point 1 above -- a serious omission, I don't consider this omission as serious).
3. Some explanation for why the computational fitting time of the finite mixture model is so long.  I understand that a few iterations of EM are necessary.  Yet, this takes 100x (?) longer than the infinite mixture model (which doesn't have EM, but must sample from the mixture at inference time?)
4. Most critically (and probably simplest to fix), the "learning" procedure for this model needs to be spelled out more clearly.  I *believe* that this is a "full Bayesian" approach (section "Bayesian update") is used, where the posterior is updated after each data point (and the "training data" is the same as "prior seen testing data").  But, I'm not sure.  It might be that the Bayesian update is only applied internally to a single sequence and that the alpha/beta parameters of the GDir prior are optimized during training (if so, how?).  Rereading the "Running Time" section, I am less certain of exactly how the training and testing are done.
5. There is no comparison to using a "normal" Dirichlet distribution (instead of the Generalized Dirichlet).  Given that the Gamma prior used for the CT models is equivalent to the Dirichlet (not the GDir) and that most of the experiments are on CT, how important are the extra parameters in the DT version?

Points 4 and 1 are the most important of those listed above.
6.

**Q7 Justification For Your Score:**

I am leaning toward "Accept," but I would like to understand better how this model is trained before rendering a final score.

**Q9 Complying With Reviewing Instructions:**

1: Yes.

---

### Decision · Program_Chairs · 2022-05-15

**Decision:**

Accept (Poster)

**Comment:**

Meta Review: The authors propose a Bayesian method for modeling trajectories over multiple states in applications where the Markov chain memorylessness assumption is unrealistic. Their solution is to create an infinite mixture of Markov chains of different lengths, such that the resulting model is not Markovian, but only requires twice the number of parameters of one chain. They introduce both discrete and continuous-time versions of their model and employ conjugate priors to ensure tractable computation. The model is examined and compared with other approaches on several open datasets.

Strength: the presentation is very clear, including the discussions on prior work; the proposed method is novel, simple, and efficient; software implementation is included to facilitate reproducibility.

Weakness: lack of comparison to semi-Markov and HMMs in the experimentation; missing references to works proposing infinite mixtures of Markov chains which should/could be addressed in the camera-ready version.

There is consensus among reviewers that the strengths overweight the weaknesses and therefore the paper should be accepted.